# Exploring Natural Language Processing through an Exemplar Using YouTube

**DOI:** 10.3390/ijerph21101357

**Published:** 2024-10-15

**Authors:** Joohyun Chung, Sangmin Song, Heesook Son

**Affiliations:** 1Elaine Marieb College of Nursing, University of Massachusetts Amherst, 224 Skinner Hall, Amherst, MA 01003, USA; joohyunchung@umass.edu; 2Department of Artificial Intelligence, Chung-Ang University, 84 Heukseok-ro, Dongjak-gu, Seoul 06974, Republic of Korea; 3Red Cross College of Nursing, Chung-Ang University, 84 Heukseok-ro, Dongjak-gu, Seoul 06974, Republic of Korea

**Keywords:** Natural Language Processing, methodology, unstructured data analysis

## Abstract

There has been a growing emphasis on data across various health-related fields, not just in nursing research, due to the increasing volume of unstructured data in electronic health records (EHRs). Natural Language Processing (NLP) provides a solution by transforming this unstructured data into structured formats, thereby facilitating valuable insights. This methodology paper explores the application of NLP in nursing, using an exemplar case study that analyzes YouTube data to investigate social phenomena among adults living alone. The methodology involves five steps: accessing data through YouTube’s API, data cleaning, preprocessing (tokenization, sentence segmentation, linguistic normalization), sentiment analysis using Python, and topic modeling. This study serves as a comprehensive guide for integrating NLP into nursing research, supplemented with digital content demonstrating each step. For successful implementation, nursing researchers must grasp the fundamental concepts and processes of NLP. The potential of NLP in nursing is significant, particularly in utilizing unstructured textual data from nursing documentation and social media. Its benefits include streamlining nursing documentation, enhancing patient communication, and improving data analysis.

## 1. Introduction

Natural Language Processing (NLP) has evolved from its foundational roots in artificial intelligence to become a critical tool in various fields, particularly in healthcare and nursing. It enhances patient care by automating the extraction of information from unstructured data in Electronic Health Records (EHRs), which improves documentation efficiency and supports real-time decision-making [1,2,3,4,5,6,7].

In recent years, nursing research and practice have increasingly shifted toward data-intensive methodologies. This change is primarily driven by the growth of unstructured data in electronic health records (EHRs), where most information is currently stored in unstructured text format [8,9]. For instance, systems such as Epic, Cerner, and Athenahealth have integrated NLP features to enhance automatic documentation and streamline data extraction from this unstructured information [10,11,12]. Research into the various applications of NLP in nursing has begun to emerge. A recent study utilized NLP to analyze a four-hour transcript from an expert panel discussion, aiming to evaluate the sustainability of a clinical trial focused on online social support interventions for Hispanic and African American caregivers of dementia patients [13]. Similarly, NLP was employed to perform qualitative analyses by comparing outcomes, time, and costs from previous studies, resulting in notable efficiency improvements [8]. Another study demonstrated the feasibility of detecting symptom groups in electronic health record (EHR) narratives, showing that an embedded NLP system outperformed traditional machine learning algorithms in identifying symptoms, including distinguishing between observed and negated symptoms, as well as medication-related side effects [4]. NLP has also been used to analyze nurse-generated data to address hospital-acquired falls by extracting relevant factors from nursing notes. In predicting hospital readmissions, NLP applied to home healthcare notes demonstrated superior performance compared to traditional models, particularly in a 30-day readmission context [14].

As NLP applications in nursing continue to expand, this evolution presents both challenges and opportunities for researchers seeking to extract meaningful insights from extensive nursing documentation, such as nurses’ notes. One significant advantage is that data-intensive methods like NLP can uncover valuable insights from nursing records, ultimately enhancing patient care and nursing practice [1,5,15,16]. NLP has proven to be a promising solution by transforming unstructured data into structured formats, which facilitates more comprehensive analysis and understanding. However, several challenges must be addressed, including concerns about data quality, ethical and privacy issues, and the need for technical expertise within the nursing field. Many nursing researchers may lack the necessary skills for data analysis and NLP, which could require additional training or collaboration with data scientists [17].

Recent articles in nursing literature highlight the application of NLP methods, providing detailed descriptions of their application in each case [1,5,15,16]. However, while many articles discuss how they analyze unstructured data briefly, few provide detailed information on the specific preprocessing steps of NLP methods. These steps are crucial because preprocessing is typically the most significant part of the NLP analysis process [18,19,20]. Preprocessing in NLP plays a pivotal role in refining raw textual data by addressing issues such as noise and spelling variations. This transformation results in a more structured format that is cleaner, consistent, and optimized for analysis and model training, thereby enhancing the overall effectiveness and efficiency of NLP applications. It often represents a substantial portion, typically ranging from 50% to 80–90%, of the total effort involved in developing an NLP pipeline. Therefore, this methodology paper offers a comprehensive overview of NLP processing in nursing. It addresses how to access existing datasets through Application Programming Interfaces (APIs), as well as the steps of preprocessing, processing, analysis, and evaluation, all using publicly available unstructured data from YouTube archives. It demonstrates the potential of NLP to extract valuable insights from data and explores its applications within nursing.

As a methodology paper, it explains the application of NLP methods through examples that utilize YouTube data to analyze social trends among middle-aged adults living independently. By employing a systematic approach that includes data access, cleaning, preprocessing, sentiment analysis, and topic modeling, it offers a robust framework for integrating NLP into nursing research to better understand the process of NLP. Each methodological step is paired with digital content designed to enhance understanding and accessibility for researchers who may be unfamiliar with these techniques. The ultimate goal is to assist nurse researchers and health researchers in effectively utilizing unstructured clinical data through NLP methods.

## 2. Background

Nursing research is rapidly becoming more data-intensive every day because the variety and volume of data within electronic health records systems (EHRs) are growing at an enormous rate. Nursing care, in particular, generates a substantial amount of unstructured data through narrative input or the use of open-ended questions, such as admission health histories, health assessment records, daily nursing notes, and nursing care plans [3,21,22]. Unlike structured data, which is organized in databases and spreadsheets, unstructured data is often textual or multimedia, making it challenging to analyze, store, and integrate.

### 2.1. Unstructured Data in Nursing

One of the primary types of documentation for nurses is nursing notes, which document patient care, treatment, and progress. While these notes are essential components of a patient’s medical record, they have not been effectively utilized for nursing research. This underutilization is largely due to a lack of awareness and technical expertise regarding the potential applications of NLP in the field [3]. In contrast, structured EHR data—such as age and diagnosis—are more frequently employed in nursing research. Unstructured data refers to information that does not conform to a standard format, such as categorical or numerical values, making it challenging to analyze, store, and integrate with other data formats. Unlike structured data, which are neatly organized in databases and spreadsheets with rows and columns, unstructured data come in various formats and is often textual or multimedia.

### 2.2. The Role of NLP

With the rise and greater adoption of data science techniques, especially in the realm of big data analytics, managing the storage and processing of unstructured data has become more feasible. Natural Language Processing (NLP), a field within computer science and artificial intelligence, has emerged as a crucial discipline that focuses on comprehending, interpreting, and processing human language in ways that yield meaningful and practical insights [17,23,24,25,26]. NLP plays a crucial role in converting unstructured data into structured data, facilitating easier analysis and processing. Therefore, NLP has the potential to extract valuable insights from unstructured textual data in document collections, offering numerous applications in nursing.

### 2.3. Historical Background of NLP

NLP has its roots in the 1950s, coinciding with the emergence of artificial intelligence (AI). Alan Turing’s work, especially his paper on “Computing Machinery and Intelligence”, was foundational for both AI and NLP [27]. Early NLP was rule-based, relying on handcrafted linguistic rules [28,29,30]. Programs like SHRDLU and ELIZA demonstrated early capabilities in understanding and generating natural language within a constrained environment [31,32].

The 1990s marked a significant era of advancement with the adoption of machine learning techniques, such as Hidden Markov Models (HMMs) for part-of-speech tagging, Named Entity Recognition (NER), [33] Support Vector Machines (SVMs), and Maximum Entropy models [34]. During the 2000s, rapid advancements in deep learning spurred significant progress in the field of NLP, including word embeddings (e.g., Word2Vec) and advanced neural networks (e.g., Long Short-term Memory (LSTM) networks and Transformers) [35,36,37,38]. Recent models, such as OpenAi’s GPT-2 and BERT, have set new benchmarks for language understanding and generation [39,40,41].

### 2.4. NLP in Nursing

NLP has become increasingly integrated into research and everyday applications, such as virtual assistants, customer service bots, and language translation services, cementing its role as an essential component of modern technology infrastructure. In nursing, several studies have utilized Natural Language Processing (NLP) for various applications. These include developing a prediction model for the risk of readmission after total joint replacement surgery [42], identifying patients with hypoglycemia for systematic reviews [2], exploring reasons for delayed nursing visits in home healthcare [1], documenting symptoms associated with a higher risk of emergency department visits in homecare patients [4], and determining fall risk factors from nurses’ notes [14]. Additionally, NLP techniques are being used for automatic documentation in Electronic Health Records (EHR) systems and for extracting data from unstructured EHRs to create structured data within these systems. Several EHR vendors, including Epic Systems, Cerner Corporation, and Athenahealth, have integrated NLP features to enhance automatic documentation and data extraction from unstructured EHRs [10,11,12].

Natural Language Processing (NLP) can greatly benefit nursing researchers and health researchers by enhancing patient care, improving communication, and optimizing data management and analysis. For instance, despite the availability of electronically accessible nurses’ notes, these notes are seldom used in nursing research. NLP can automate the extraction and summarization of critical information from these notes.

### 2.5. Future Directions of NLP in Nursing

As nursing research evolves, the integration of NLP holds great promise for enhancing the utility of unstructured data, improving patient outcomes, and optimizing nursing practice. By leveraging advanced NLP techniques, nurse researchers can unlock valuable insights that have the potential to transform healthcare delivery [7]. In nursing practice, NLP can greatly reduce the time spent on documentation by automating transcription from speech and various sources of EHR data. This efficiency allows nurses to dedicate more time to direct patient care. NLP provides real-time decision support by analyzing patient records and medical literature, enabling nurses to make informed decisions and alerting them to potential issues, such as drug interactions. NLP can be seamlessly integrated into personal health assistants, such as Siri or Google Assistant, providing significant benefits for older adults in community settings. This technology allows users to track health metrics, such as blood pressure, and receive medication reminders using simple voice commands. Users can inquire about symptoms or seek dietary advice, although it is essential to clarify that such information should not replace professional medical advice. Furthermore, NLP can enhance continuity of care by connecting users with relevant nursing resources and support. In nursing applications, NLP can include features such as journaling for diabetes management and reminders for individuals with mild cognitive impairment living independently. Improving medication management helps users adhere to their treatment plans and enhances overall health.

Specifically, in in-home care settings, NLP holds significant potential for understanding the nuanced challenges faced by older adults. By transcribing and analyzing spoken interactions, researchers can gain insights into emotional well-being, communication barriers, and specific care needs, as well as identify patterns of anxiety or loneliness. Perhaps topic modeling can reveal recurring themes related to daily living and social support. These insights can inform caregivers and older adults about critical areas that need attention, ultimately leading to improved care strategies that cater to the unique experiences of older individuals.

Additionally, NLP can be specifically tailored for mental health support through chatbots, ensuring privacy and anonymity. These chatbots can conduct initial assessments to identify individuals’ mental health needs and direct them to appropriate local resources, such as support groups and hotlines. The implications of NLP in nursing span multiple areas, influencing various aspects of patient care, research, and operational efficiency within nursing practice.

## 3. Materials and Methods

### 3.1. Overview of the NLP Methodology Utilizing an Exemplar

In this methodology paper, we will examine the NLP approach through an exemplar. NLP involves the conversion of unstructured human language data into structured data that can be understood by machines [26]. This paper presents an exemplar that demonstrates how to perform NLP using Python, specifically examining the health-related social dynamics experienced by middle-aged adults living alone. This example was selected for three reasons. First, it utilizes publicly available data from YouTube on general health and nursing topics, without requiring specialized knowledge. The study specifically examines how middle-aged adults engage with YouTube and utilize the platform for health-related content. Second, this case enhances understanding of the steps involved in NLP and raises awareness of its applications. Finally, by utilizing real-world data, it clearly illustrates the technical processes in NLP, enabling other researchers to replicate the study and deepen their insights. On YouTube, individuals, content creators, and organizations can upload a wide array of video content encompassing various topics, including entertainment, education, news, music, and tutorials. Users can access YouTube through websites or mobile applications and engage with videos by liking, commenting, subscribing to channels, and sharing content with others. With its extensive collection encompassing millions of videos, YouTube has emerged as a significant platform for entertainment, information, and community involvement. By utilizing YouTube, the example study aimed to explore the health-related social dynamics faced by middle-aged adults living alone in single households, focusing specifically on how they interact with and utilize the platform. This approach helps us understand the unique challenges and resources available to this demographic, highlighting the importance of social connections and support systems in their lives. Existing research has identified the negative consequences associated with living alone, including reduced quality of life [43,44,45], increased risk of depression [46,47], increased prevalence of chronic diseases [46], psychological distress, and cognitive decline [48]. Furthermore, living alone has been linked to an unhealthy lifestyle, characterized by regular consumption of processed foods [49], smoking [50], physical inactivity [50,51], and alcohol consumption [52]. However, these findings do not provide a comprehensive understanding of the range of health-related social dynamics that middle-aged adults living alone may encounter. Therefore, we chose YouTube as the focal point of this study, as it is a prominent platform for sharing and viewing videos.

The methodology consists of five key steps: accessing data through YouTube’s API, cleaning the data, preprocessing it (including tokenization, sentence segmentation, and linguistic normalization), conducting sentiment analysis using Python, and performing topic modeling (Figure 1). These steps collectively enable the analysis and extraction of meaningful information from text data, allowing machines to perform tasks, such as language translation, sentiment analysis, information retrieval, and text summarization. In addition, we would like to acknowledge that ChatGPT was employed to capture historical events and ideas pertinent to the applications of Natural Language Processing (NLP) in nursing and healthcare research from 1950 to 2000.

### 3.2. Phase 0—Access Existing Data Using Application Programming Interfaces (APIs)

In data science, data preprocessing involves actions taken to obtain the dataset and convert raw data into a well-organized and structured format, suitable for subsequent analysis and modeling tasks. First, existing data for research can be accessed through several methods, depending on the nature of the data and its availability. Various data sources utilize APIs, which are protocols that enable users to query resources and retrieve data in a machine-readable format. Researchers often employ APIs to download text collections, such as scholarly journal articles, facilitating automated text mining on the obtained corpus. In this case, we will demonstrate the process of accessing the YouTube API. To access YouTube data for research purposes, the recommended approach is to utilize the YouTube API.

The YouTube API enables developers to interact with YouTube’s features, including retrieving data related to videos, channels, playlists, and comments. The process is outlined below.

Create a Project and Obtain API Key: To utilize the YouTube API, you need to create a project in the Google Cloud Console and enable the YouTube Data API for that project. This will provide you with an API key, which you can use to authenticate your requests.Set Up API Client: Depending on your programming language, you will need to set up an API client or library that can make requests to the YouTube API. Popular options include Python (using libraries like google-api-python-client), JavaScript, or other languages supported by YouTube API.Authenticate and Make API Requests: You will need to authenticate your API requests using the API key obtained in Step 1. This involves including the API key in your requests to identify and authorize access to the YouTube API.Retrieve Data: Once authenticated, you can begin making API requests to retrieve specific data from YouTube. For example, you can fetch information about videos, such as their titles, descriptions, view counts, or comments. We retrieved channel information and searched for videos based on keywords (middle-aged in a single household).

It is important to note that when accessing existing data for research, researchers need to comply with ethical guidelines, privacy regulations, and any specific data use agreements or restrictions associated with the data source.

Exemplar

We aimed to analyze comments from YouTube videos on the health-related social dynamics experienced by middle-aged adults residing in single households. Here is a guide on how to do it, using a tool called Colab and a technique called Topic Modeling (refer to the codes and detailed annotation—Appendix A and videos). We chose 30 channels, specifically focusing on middle-aged individuals living in single households. Our selection criteria were based on keywords related to this demographic. We collected 2736 videos, along with their corresponding comments, descriptions, and transcripts, and compiled all this information into a CSV file.

### 3.3. Phase 1: Preprocessing in NLP

In data science, data preprocessing refers to the process of identifying, correcting, and removing errors, inconsistencies, and inaccuracies from a dataset. It involves detecting and handling various types of issues present in the data to ensure its quality and reliability for analysis and modeling. In the context of Natural Language Processing (NLP), preprocessing refers to the set of tasks and techniques applied to textual data before it can be employed for analysis or modeling [53].

Removing Special Characters and Numbers: Discarding special characters, symbols, and numeric values that may not be relevant for analysis or may introduce noise into the data.Thus, special characters (e.g., _,_\n) and emoticons were removed from the data.Non-relevant links were removed (e.g., advertisement links).Lowercasing: Converting all text to lowercase to ensure consistency and reduce the complexity of text analysis. This prevents issues with different cases of the same word being treated as separate entities.Tokenization: Breaking the text down into individual tokens, such as words, sentences, or subwords. Tokenization is the fundamental step in NLP, as it forms the basis for subsequent analysis.Stop Word Removal: Eliminating common words (e.g., “and”, “the”, “is”) known as stop words that do not carry significant meaning and are often irrelevant for analysis. This helps reduce noise in the text data.Punctuation Removal: Stripping out punctuation marks like periods, commas, and quotation marks, as they generally do not contribute to the overall meaning of the text.Lemmatization/Stemming: Reducing words to their base or root form. Lemmatization aims to transform words to their dictionary form (lemma), whereas stemming involves reducing words to their common stem. These techniques help consolidate words with similar meanings and reduce vocabulary size.

Preprocessing is crucial in NLP, as it aids in the transformation of unstructured text data into a more structured and manageable format for analysis. These preprocessing steps ensure that the text is in a suitable form for extracting meaningful insights, building models, and applying various NLP techniques. In our specific example, we applied name masking to YouTube names and performed text concatenation [53].

YouTube Names Masking: YouTube channel names are masked to a common identifier.Text Concatenation: Video titles and comments are combined for further processing (i.e., [title; comment]).

### 3.4. Phase 2: Text Representation

This step involves converting preprocessed text into a numerical representation, known as “text representation”(Figure 2). It is a crucial phase in NLP, where we transform the preprocessed text into a numerical format that machine learning algorithms can comprehend and process. Common approaches include techniques like bag-of-words (BoW), Term frequency-inverse document frequency (TF-IDF), word embeddings (e.g., Word2Vec, GloVe), or deep learning-based representations (e.g., BERT, GPT).

First, BoW represents text as a collection of unique words in a document, disregarding grammar and word order. BoW is simple and efficient but does not capture word semantics or context. Second, a common approach is the term frequency-inverse document frequency (TF-IDF). TF-IDF considers the importance of each word in a document relative to a collection of documents. It measures the frequency of a term in a document (TF) but discounts it by the frequency of the term across all documents (IDF). TF-IDF helps identify important words that are not common across all documents. Word embeddings turn words into dense vectors in a continuous space, making similar words have similar vectors. Third, word embeddings capture the meanings and relationships between words, helping us find similarities between words, phrases, or documents. These are known as semantic relationships. Last, deep learning-based representations include BERT (Bidirectional Encoder Representations from Transformers) and GPT (Generative Pre-trained Transformer). These models learn the meanings of words and sentences by training on vast amounts of text data. Similar to ChatGPT, they capture complex relationships and nuances in language, achieving state-of-the-art performance in various NLP tasks.

The obtained corpus is loaded for model creation using the Python Pandas library.

Initiating the BERTopic Model: To create the model using BERTopic, you first load the corpus as a list and then pass it to the fit_transform method. This method accomplishes the following steps: it fits the model to the corpus, subsequently generates topics, and returns results containing those topics [54].Visualization: For visualization purposes, we utilized the intertopic distance map and barchart methods. First, with the intertopic distance map method, you can represent the topic sizes and items generated by corresponding words (Figure 3). Next, using the barchart method, we created bar charts based on c-TF-IDF scores to display the words that constitute selected topics. This allows you to compare the scores across topics and examine the results generated for the topics (Figure 4).

### 3.5. Phrase 3: Feature Extraction

The third phase is feature extraction, which involves identifying key features from text to capture important characteristics or patterns. Common techniques include n-grams, which find sequences of N consecutive words to analyze patterns or relationships; part-of-speech (POS) tagging, which assigns grammatical categories like nouns, verbs, and adjectives to words in a sentence to understand their roles; named entity recognition (NER), which identifies and classifies specific names such as people, organizations, and locations; syntactic parsing, which examines the grammatical structure of sentences to understand how words relate to each other; and topic modeling, which uncovers topics or themes within a collection of documents by looking at how words often appear together. By applying these techniques, one can better understand the underlying structure and meaning of the text, facilitating tasks such as information retrieval, sentiment analysis, or text summarization [54].

### 3.6. Phrases 4 & 5: Model Training and Evaluation

Develop and train machine learning or deep learning models using the preprocessed text data and extracted features. This typically involves splitting the data into training, validation, and test sets, selecting appropriate algorithms or models, tuning hyperparameters, and evaluating the model’s performance using suitable metrics (Appendix A).

Postprocessing and Visualization: After running the NLP models, refine the output to generate meaningful insights. This can involve additional data analysis techniques, visualization methods, or summarization techniques to present the findings clearly and understandably.Iteration and Improvement: Continuously refine and improve the NLP models and analysis based on feedback, evaluation results, and specific requirements of the domain. This may include adjusting preprocessing steps, modifying features, exploring different algorithms, or incorporating domain knowledge.

#### Results from the Exemplar

As shown in Figure 5 and Figure 6, when the number of topics in the left panel is 23, there is no overlap between the circles representing topic-lexical item distributions. This indicates a clear separation between the topics, suggesting effective distinction in high-dimensional space, which enhances the model’s training performance on the training set. In contrast, a poorly performing topic model may show many small, overlapping bubbles clustered in one area of the chart [53].

In this example, for men, topics 1, 4, 6, 19, and 22 show a strong relationship, indicating they frequently co-occur, while topics that are farther apart reflect fundamentally different issues. The right side of the chart displays associated topic probability values, highlighting the prevalence of each topic, with higher probabilities indicating greater relevance in male discourse. For instance, Topic 1 accounts for approximately 53% of the tokens in the text document for men, while for women, Topic 1 represents about 15% of the tokens.

The top five topics were identified for two distinct groups living alone: (1) by gender and (2) by generation (those under 30 years old and those aged 30 and older). Table 1 presents the top five topics analyzed by gender and generation. Males predominantly discuss topics related to personal challenges and lifestyle, with “Divorce” and “unemployment” appearing frequently across multiple topics. For instance, Topic 1 focuses on the struggles of “divorce”, “unemployment”, and the experience of being an “old bachelor”. This suggests a shared concern among middle-aged men regarding societal perceptions and personal hardships related to these issues. In contrast, females engage with topics that center around social identity and daily experiences. Topic 1 highlights themes of “old maid”, “single”, and “weekend”, indicating a focus on the social implications of being unmarried. The recurring mention of “Mukbang” (eating broadcasts) across several topics underscores a cultural trend among women that blends food, social connection, and personal narratives.

When examining generational differences, the MZ generation (under 30 years old) shows a strong inclination towards themes of leisure and self-presentation, as seen in Topic 1, which includes “saving”, “camping”, and “Mukbang”. This reflects a lifestyle that values experiences, such as camping and online content creation. Conversely, the older generation focuses more on existential themes related to loneliness and daily life. Topic 1 addresses the life of an “old maid” in her 40s, emphasizing daily experiences and the challenges of being unmarried. This generational divide highlights differing priorities, with younger individuals leaning towards social activities and older individuals reflecting on personal circumstances and social expectations.

Unfortunately, there are no topics related to health and medical concerns or any associated activities, regardless of gender or generation. This may be due to limited data and resources specifically addressing health-related issues. Additionally, health topics may be less prominent on YouTube due to privacy concerns and the sensitivity of the information, even though such content can be shared anonymously on the platform.

## 4. Conclusions

We have explored the application of NLP in research using an exemplar, highlighting its value as a tool for extracting important insights from large text datasets. In conclusion, the analysis in Figure 4 and Figure 5 reveals a clear separation between topics when set to 23, indicating the model’s strong ability to distinguish themes in high-dimensional space. For male participants, key topics like “divorce” and “unemployment” frequently co-occur, reflecting shared concerns about societal perceptions and personal struggles. In contrast, female participants focus on social identity and daily experiences, with mentions of “Mukbang” highlighting cultural connections. Generational differences also emerge younger individuals (under 30) prioritize themes of leisure and self-presentation, while older participants reflect on loneliness and the challenges of being unmarried. This highlights the differing interests across age groups. Notably, there is a lack of topics related to health and medical concerns, likely due to privacy issues and the sensitivity of such topics on platforms like YouTube. Overall, these findings underscore the need for further exploration of underrepresented themes, particularly in health discourse, while highlighting the distinct concerns of different demographic groups.

Our illustrative example serves as a valuable demonstration of how to understand the NLP method step by step, using YouTube comments. NLP streamlines the tasks of data collection and extraction through automation. Furthermore, NLP empowers researchers to automatically classify and group text data into topics or thematic categories.

Traditional research methods, which include both qualitative and quantitative approaches, have distinct advantages and limitations when compared to NLP. Qualitative research provides in-depth insights into participants’ experiences and contexts through techniques like focus groups and interviews. In contrast, quantitative research identifies patterns across larger populations, allowing for generalizable findings. However, traditional research can be time-consuming, and qualitative findings are often subject to researcher bias and limited in their generalizability. On the other hand, NLP excels in automating the analysis of vast amounts of unstructured data, offering scalability and real-time insights that traditional methods may struggle to provide.

In terms of clinical implications, incorporating NLP into nursing research can significantly enhance patient care. For example, NLP algorithms can analyze electronic health records (EHRs) to identify patients at high risk for conditions such as sepsis by examining trends in vital signs and lab results. By alerting nurses to these risks, timely interventions can be initiated, potentially improving patient outcomes based on insights drawn from vast amounts of unstructured data. Moreover, NLP can streamline documentation processes by automatically transcribing and organizing clinical notes. This efficiency allows nurses to dedicate more time to patient interactions rather than paperwork. As a result, increased direct patient care can strengthen the nurse–patient relationship, leading to higher patient satisfaction and improved adherence to treatment plans. In summary, integrating NLP into nursing research not only offers deeper insight into patient experiences but also translates into actionable strategies for enhancing care and outcomes.

## Figures and Tables

**Figure 1 ijerph-21-01357-f001:**
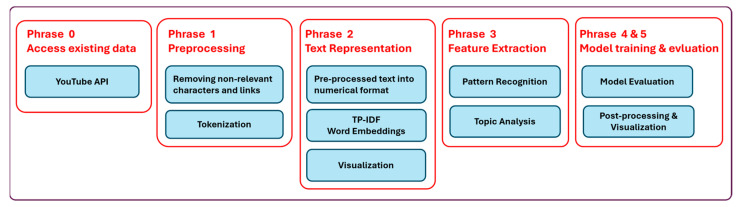
The five steps of NLP.

**Figure 2 ijerph-21-01357-f002:**
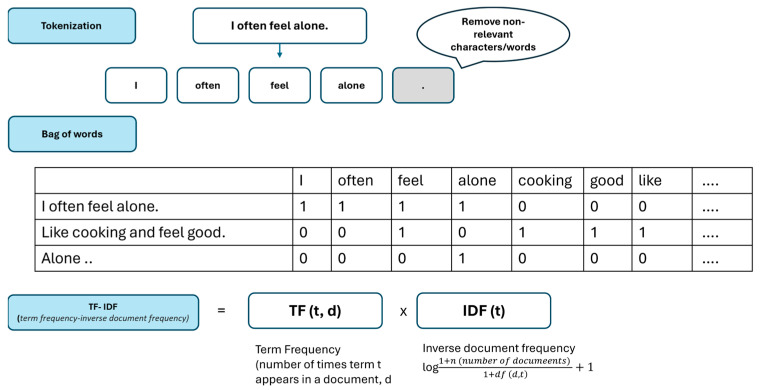
Conceptualization of Text representation.

**Figure 3 ijerph-21-01357-f003:**
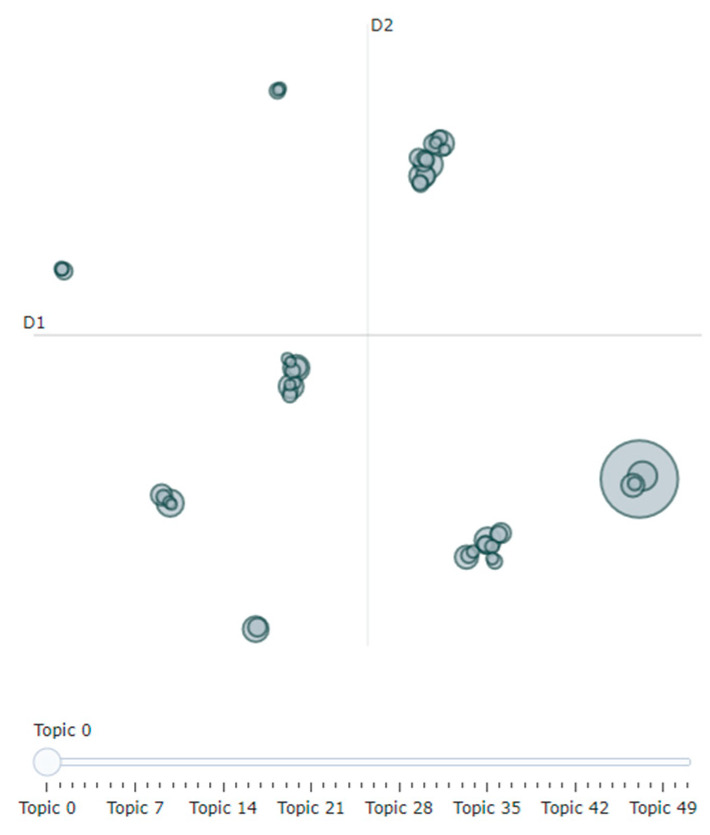
Visualization–Intertopic Distance Map.

**Figure 4 ijerph-21-01357-f004:**
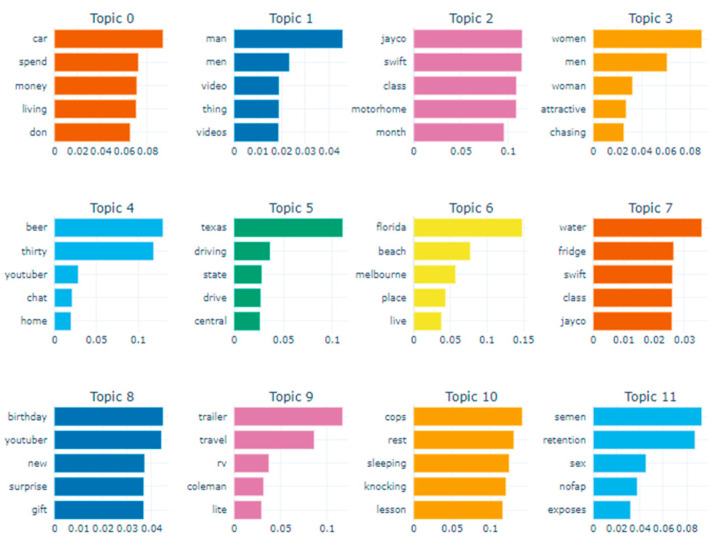
Visualization–Bar charts of selected topics.

**Figure 5 ijerph-21-01357-f005:**
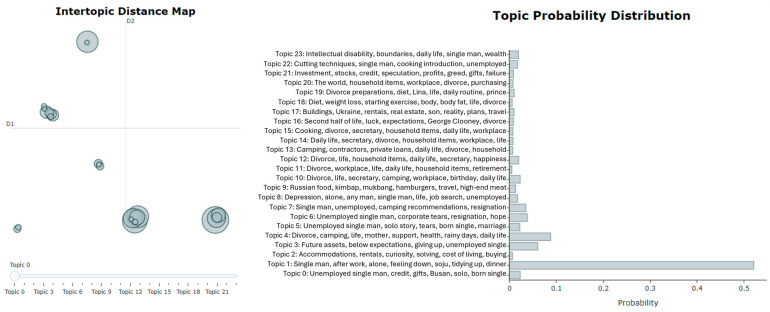
Intertopic distance map and topic probability for males.

**Figure 6 ijerph-21-01357-f006:**
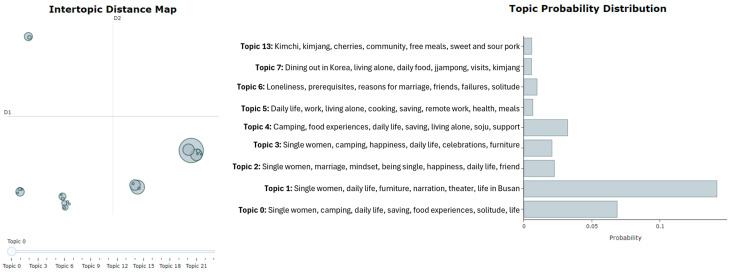
Intertopic distance map and topic probability for females.

**Table 1 ijerph-21-01357-t001:** The five steps of NLP.

Categories			Those Who Were Observed in Korean
Gender	Male	Topic 1	Divorce, unemployment, old bachelor, daily life, work
		Topic 2	Divorce, old bachelor, life, readiness, vacation
		Topic 3	unemployment, old bachelor, credit, truth, no jobs
		Topic 4	Lodging, travel, rent, question, solution
		Topic 5	Lodging, travel, rent, question, solution
	Female	Topic 1	Old maid, single, alone, weekend, marriage
		Topic 2	Old maid, single, Mukbang (eatcasting), daily life, alone
		Topic 3	Simple, Mukbang (eatcasting), daily life, saving, living alone
		Topic 4	Seoul, old maid, single, narration, daily life
		Topic 5	Mukbang (eatcasting), dinner, single, old, maid, saving
Generation	MZ	Topic 1	Saving, camping, tent, Mukbang (eatcasting), V Log, appearance
		Topic 2	Saving, Mukbang(eatcasting), seeing, V log, appearance
		Topic 3	Camping, sleeping in car, Mukbang(eatcasting), Soju (one type of alcohol), solo
		Topic 4	Old maid, 30s, old bachelor, drinking alone, single
		Topic 5	Old maid, drinking alone, boss, unemployment, V log
	Old	Topic 1	Old maid in her 40s, single, vlog, unmarried, drinking alone, daily life, good life alone
		Topic 2	Unemployed, old bachelor’s daily vlog, unmarried life in his 40s, old maid, Seoul season
		Topic 3	Married, unmarried, alone, not in 40s, daily life, reasons, friends, vlog
		Topic 4	Vlog, Old bachelor in his 40s, Daily life, Divorce Life, Old maid, Unmarried, Unemployed Person Furniture
		Topic 5	Food, Cooking, Living, Vlog, Daily Life, Savings, Simple Diet, Health

## Data Availability

The data, including YouTube comments stored in CSV format, was loaded from a specified drive path. Comments and titles were further cleaned, masked, and processed. Quantiles were computed for length-based filtering, and the dataset was sampled and concatenated. The final Data frame was used to combine titles and comments.

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
