# Peer review of "Exploring Natural Language Processing through an Exemplar Using YouTube"

_ijerph, 2024, doi:10.3390/ijerph21101357_

Round 1

Reviewer 1 Report

Comments and Suggestions for Authors

Dear authors,

Thank you for this very interesting and relevant topic. 

Here are my comments for improvement for your consideration.

1. Introduction and background. 

The introduction should be able to present the thesis statement, followed by the explanation of the points and the evidence and examples that support the explanation and the points.  These were well presented in the background section of the paper.

The background should be able to discuss the specifics of the study such as the gaps that lead to the conduct of the study and other more focused details. These were addressed in the introduction section of the paper. The narration of the technology evolution in the background section which is comprehensive overshadowed the purpose of the study discussed in the introduction part.

I suggest for you to re organize the introduction and background to direct the readers on what the study is all about. 

Materials and Methods

The process of NLP was well outlined and explained in this section, however, as this is a case study a case study investigating the 153 health-related social dynamics experienced by middle-aged adults living in single 154 households, I was not able to appreciate how the NLP was  applied in this context. The methodology section is very technical and was not able to contextualized the process with the case presented.

It would be nice to show some images showing the specific application of the steps to make it better understood and for easier following.

I have not seen the result of the case study. What was generated from the NLP from step 0 to the final output?

How is this case study related to nursing research as implied from the title of the article?

I suggest for you to contextualize the methods section as well as to add a results section and present the output from the NLP process from the case presented.

Thank you and good luck.

Author Response

Reviewer 1

 Introduction and background. 

The introduction should be able to present the thesis statement, followed by the explanation of the points and the evidence and examples that support the explanation and the points.  These were well presented in the background section of the paper.

The background should be able to discuss the specifics of the study such as the gaps that lead to the conduct of the study and other more focused details. These were addressed in the introduction section of the paper. The narration of the technology evolution in the background section which is comprehensive overshadowed the purpose of the study discussed in the introduction part.

I suggest for you to re organize the introduction and background to direct the readers on what the study is all about. 

Response
Thank you for your comment. We have revised the introduction and background to present a clearer thesis statement that highlights the gaps leading to this study, making it more focused overall.

Materials and Methods

The process of NLP was well outlined and explained in this section, however, as this is a case study a case study investigating the 153 health-related social dynamics experienced by middle-aged adults living in single 154 households, I was not able to appreciate how the NLP was  applied in this context. The methodology section is very technical and was not able to contextualized the process with the case presented.

It would be nice to show some images showing the specific application of the steps to make it better understood and for easier following.

Response

Thank you for your feedback. I appreciate your insights on the application of NLP in the context of our case study. I understand your concerns about the methodology section and its technical nature. I will work on contextualizing the NLP process more clearly in relation to the specific dynamics of middle-aged adults living alone, why we chose this and how the NLP was applied in this context. Including images to illustrate the application of the steps is a great idea. We added the figure 1 for the steps of NLP.  

I have not seen the result of the case study. What was generated from the NLP from step 0 to the final output?

Response
Thank you for the feedback. We have included the results of the case study.

How is this case study related to nursing research as implied from the title of the article?

I suggest for you to contextualize the methods section as well as to add a results section and present the output from the NLP process from the case presented.

Response
Thank you very much for the feedback. We’ve added contextual information in the methods section to clarify how the case study relates to nursing research. For instance, we included details on how the case study was developed as an illustrative example in nursing, along with a brief discussion of the gap it addresses in the current literature.

Reviewer 2 Report

Comments and Suggestions for Authors

Thank you for giving me the opportunity to review your paper. Your research represents a novel approach to nursing research and is highly intriguing. The paper is well-written and clearly presents the research. However, to further enhance the paper's quality, I suggest the following revisions.

This study uses YouTube to survey single-elderly individuals. I believe this approach can be further developed. For example, could researchers conduct video observations in the homes of older individuals and analyze the footage using NLP to uncover new insights into the challenges of home care? This approach could provide a more detailed and authentic view of daily life. I suggest highlighting this potential as a major contribution of NLP to nursing research.

Introduction

The introduction offers a thorough overview of NLP. However, the discussion of NLP in nursing is somewhat superficial, merely listing relevant papers. The paper should provide more specific examples of how NLP is being used in nursing practice and the challenges faced. While the NLP background is detailed, it could be more concise. From line 125, please expand on specific applications, results, and future directions for NLP in nursing.

Conclusion

To enhance the impact of this research on the nursing field, the following points should be addressed:  While the potential of NLP is evident, the paper could benefit from a more explicit discussion of how NLP can provide unique insights into nursing research questions. Please compare NLP to traditional methods and highlight its specific advantages.  Finally, it would be beneficial to discuss the potential clinical implications of using NLP in nursing research, such as improving patient care or outcomes.

Author Response

Reviewer 2

Thank you for giving me the opportunity to review your paper. Your research represents a novel approach to nursing research and is highly intriguing. The paper is well-written and clearly presents the research. However, to further enhance the paper's quality, I suggest the following revisions.

This study uses YouTube to survey single-elderly individuals. I believe this approach can be further developed. For example, could researchers conduct video observations in the homes of older individuals and analyze the footage using NLP to uncover new insights into the challenges of home care? This approach could provide a more detailed and authentic view of daily life. I suggest highlighting this potential as a major contribution of NLP to nursing research.

 Response

Thank you very much for your valuable feedback. We’ve added more discussion on that in background (specifically 2.5). Also, we’ve included more discussion of NLP in nursing including specific articles in current literature. We added specific application, results and futrue directions for NLP in ursing (both in introduction and background).

Introduction

The introduction offers a thorough overview of NLP. However, the discussion of NLP in nursing is somewhat superficial, merely listing relevant papers. The paper should provide more specific examples of how NLP is being used in nursing practice and the challenges faced. While the NLP background is detailed, it could be more concise. From line 125, please expand on specific applications, results, and future directions for NLP in nursing.

 Response

Thank you for your feedback. We’ve included more discussion of NLP in nursing including specific articles in current literature. We added specific application, results and futrue directions for NLP in ursing (both in introduction and background).

Conclusion

To enhance the impact of this research on the nursing field, the following points should be addressed:  While the potential of NLP is evident, the paper could benefit from a more explicit discussion of how NLP can provide unique insights into nursing research questions. Please compare NLP to traditional methods and highlight its specific advantages.  Finally, it would be beneficial to discuss the potential clinical implications of using NLP in nursing research, such as improving patient care or outcomes.

Response

Thank you for your insightful feedback. We appreciate the opportunity to clarify how NLP can provide unique insights into nursing research questions.

Round 2

Reviewer 1 Report

Comments and Suggestions for Authors

Dear Authors,

Thank you for your revised and improved paper. It is now a more complete version.

I have seen the effort to improve the paper, however, there are still some points that need reconciliation and improvement.

1. Title - The title specifically mentioned NLP in Nursing Research. Although you have tried to expound on how NLP can be a significant approach in Nursing research, the current research is not nursing. Aside from using you tube which has no relation to nursing, the NLP result, moreover, did not show any nursing aspect. I suggest to rephrase or to remove the words nursing research in the title and replace it with the specific case under study. What is the specific case studied using the NLP approach? Make it clear in the title.

2. Methods

The description of the different steps/phases of NLP is very technical and can be searched from other sources. Aside from you  not including references in your methodology (Unless these phases came from you), the steps are very generic. I would like to reiterate for you to contextualize the description of the phases and support with images (after  for the reader to have a better understanding and for researchers to learn how the phases are done reading from your study. 

3. Result- It is commendable that this section has been added as it completed the picture; however I find it too short. After going through the rigorous phases of NLP, I find the result generated quite lacking in depth. A new researcher who will attempt to use NLP may find the process cumbersome and not commensurate to the result generated.

The title of Table 1 I think should not be phases, but these were the themes generated from the NLP.

4. Conclusion- You did not conclude about the case study. You focused on the methodology. You should be able to conclude on both aspects.

If the focus of your study is to showcase NLP as a design, then your study is not a case study but is more of a methodological study. 

You have to decide whether to focus on the case at hand or the methodology so that from the title to the conclusion you will be more consistent. Based on the presentation of your paper, it is more of a methodological study than a case study.

In this case you can frame your title more appropriately.

Your patience in further organizing your paper will be significant to readers and  researchers who would like to use NLP approach.

Thank you and good luck.

Author Response

Comments

I have seen the effort to improve the paper, however, there are still some points that need reconciliation and improvement.

  1. Title - The title specifically mentioned NLP in Nursing Research. Although you have tried to expound on how NLP can be a significant approach in Nursing research, the current research is not nursing. Aside from using you tube which has no relation to nursing, the NLP result, moreover, did not show any nursing aspect. I suggest to rephrase or to remove the words nursing research in the title and replace it with the specific case under study. What is the specific case studied using the NLP approach? Make it clear in the title.

- Thank you for your feedback regarding the title and focus of my research. I appreciate your insights on the importance of aligning the title with the specific case under study. We rephrased the title to better reflect the specific application of NLP in the context of the research.

  1. Methods

The description of the different steps/phases of NLP is very technical and can be searched from other sources. Aside from you not including references in your methodology (Unless these phases came from you), the steps are very generic. I would like to reiterate for you to contextualize the description of the phases and support with images (after for the reader to have a better understanding and for researchers to learn how the phases are done reading from your study. 

- Thank you for your feedback. We also created a video and aimed to capture still images from it to accompany each phrase. This will help readers better understand how the phrases are developed through the study.

  1. Result- It is commendable that this section has been added as it completed the picture; however I find it too short. After going through the rigorous phases of NLP, I find the result generated quite lacking in depth. A new researcher who will attempt to use NLP may find the process cumbersome and not commensurate to the result generated.

The title of Table 1 I think should not be phases, but these were the themes generated from the NLP.

- Thank you for your feedback regarding the Results section. I appreciate your recognition of its importance in providing a comprehensive overview. We tried to include more detailed interpretations and implications of the findings to better support new researchers in understanding the process.

  1. Conclusion- You did not conclude about the case study. You focused on the methodology. You should be able to conclude on both aspects.

If the focus of your study is to showcase NLP as a design, then your study is not a case study but is more of a methodological study. 

You have to decide whether to focus on the case at hand or the methodology so that from the title to the conclusion you will be more consistent. Based on the presentation of your paper, it is more of a methodological study than a case study.

In this case, you can frame your title more appropriately.

Your patience in further organizing your paper will be significant to readers and researchers who would like to use NLP approach.

Thank you and good luck.

Thank you for your feedback. I appreciate your insights regarding the need for a more balanced conclusion that addresses both the case study and the methodology. We revised the conclusion to clearly summarize the findings from the case study while also discussing the methodological aspects of using NLP. We understand your point about the study's focus. We revised the title to reflect that focus more accurately. We believe this helps ensure consistency throughout the paper, from the title to the conclusion.

Thank you for all, again.

Reviewer 2 Report

Comments and Suggestions for Authors

Thank you for your thorough revision of the manuscript. The improvements you've made have significantly enhanced the overall quality of the paper. However, we have noticed a few minor inconsistencies in the citation style. Please revise all citations accordingly to ensure full compliance with our journal's guidelines.

Author Response

Reviewer 2

Thank you for your thorough revision of the manuscript. The improvements you've made have significantly enhanced the overall quality of the paper. However, we have noticed a few minor inconsistencies in the citation style. Please revise all citations accordingly to ensure full compliance with our journal's guidelines.

  • Thank you for your kind words and for reviewing the manuscript. I appreciate your feedback on the citation style. I carefully revised all citations to ensure they comply with the journal's guidelines.